# Interatomic Potential to Predict the Favored Glass-Formation Compositions and Local Atomic Arrangements of Ternary Al-Ni-Ti Metallic Glasses

Qilin Yang, Jiahao Li, Wensheng Lai, Jianbo Liu * and Baixin Liu

Key Laboratory of Advanced Materials (MOE), School of Materials Science and Engineering, Tsinghua University, Beijing 100084, China; yangql16@mails.tsinghua.edu.cn (Q.Y.); lijiahao@mail.tsinghua.edu.cn (J.L.); wslai@mail.tsinghua.edu.cn (W.L.); dmslbx@tsinghua.edu.cn (B.L.)
* Correspondence: jbliu@tsinghua.edu.cn

**Abstract:** An empirical potential under the formalism of second-moment approximation of tight-binding potential is constructed for an Al-Ni-Ti ternary system and proven reliable in reproducing the physical properties of pure elements and their various compounds. Based on the constructed potential, molecular dynamic simulations are employed to study metallic glass formations and their local atomic arrangements. First, a glass-formation range is determined by comparing the stability of solid solutions and their corresponding counterparts, reflecting the possible composition region energetically favored for the formation of amorphous phases. Second, a favored glass-formation composition subregion around $Al_{0.05}Ni_{0.35}Ti_{0.60}$ is determined by calculating the amorphous driving forces from crystalline-to-amorphous transition. Moreover, various structural analysis methods are used to characterize the local atomic arrangements of $Al_{0.05}Ni_xTi_{0.95-x}$ metallic glasses. We find that the amorphous driving force is positively correlated with glass-formation ability. It is worth noting that the addition of Ni significantly increases the amorphous driving force configurations of fivefold symmetry and structural disorder in $Al_{0.05}Ni_xTi_{0.95-x}$ metallic glasses until the content of Ni reaches approximately 35 at%.

**Keywords:** interatomic potential; metallic glasses; glass formation; atomic arrangements



## 1. Introduction

Metallic glasses (MGs) are expected to be excellent engineering and functional materials due to their high strength, high hardness, large elastic elongation, good corrosion resistance and soft magnetic behavior [1–3]. Since Paul Duwez first obtained metallic glass ($Au_{75}Si_{25}$) by liquid melt quenching in 1960 [4], the preparation, formation, structure and property investigation of metallic glasses have attracted considerable attention. However, the complexity of manufacturing large-size MGs has always been the main limitation to their application. To develop MGs with high glass-formation ability (GFA), several empirical criteria or rules have been proposed to predict the glass-formation range (GFR) and GFA. In experiments, the GFA of MGs is evaluated proportionally to either their critical cooling rate or casting size [5]. To predict MG formation, Turnbull proposed a deep eutectic criterion [6], Liu proposed a structural difference rule [7], Egami proposed the atomic size effect on the formability of MG [8] and Inoue formulated three basic empirical rules [9–11]. A method to predict metallic glass formation using the interatomic potential via relevant atomistic simulation has been proposed. Indeed, interatomic potentials have been proven valuable for prediction of the GFR and GFA of metal systems [12–14].

In recent years, the binary and ternary metallic glasses of Al-Ni-Ti systems have been intensively studied experimentally in terms of their structures, properties and metallic glass formation [15,16]. For instance, Howie explored metallic glass formation in Al-Ni-Ti and demonstrated that fwhm and corrosion resistance are correlated and that a large

fwhm, attributed to a glassy phase, is necessary for the highest corrosion resistance [15]. In addition, numerous experimental studies have shown that Al-rich MGs exhibit ultrahigh strength and plasticity [17] and excellent corrosion resistance properties [18], Ni-based MGs exhibit relatively high GFA [19] and wear resistivity [20] and Ti-based MGs have been widely used as outstanding lightweight structural materials [21–23]. The addition of Al and Ni can improve the GFA of Ti-based MGs [24,25]. Therefore, it would be intriguing to design favored glass-formation compositions for ternary Al-Ni-Ti metallic glasses and provide guidelines for experimentation.

In the present work, we focused our attention on an Al-Ni-Ti system and directly predicted the metallic glass formation of Al-Ni-Ti alloys according to their interatomic potential. We constructed a new interatomic potential for the investigated Al-Ni-Ti system, predicted the GFR and GFA using molecular dynamics (MD) simulations combined with Monte Carlo (MC) simulations and investigated the atomic arrangements associated with metallic glass formation.

## 2. Model and Methods

### 2.1. Potential Formalism

In atomistic simulations, the interatomic potential plays a dominant role in describing the realistic interatomic interactions of a system. In the present work, the TB-SMA potential [26] was adopted, which has been proven relevant in containing and distinguishing the face-centered cubic (fcc) and hexagonal close-packed (hcp) structures in Al-Ni-Ti metal systems. The total energy is expressed as:

$$E = \sum_i \left\{ \frac{1}{2} \sum_{j \neq i} \phi(r_{ij}) - \sqrt{\sum_{j \neq i} \psi(r_{ij})} \right\}, \tag{1}$$

where $r_{ij}$ is the distance between atoms $i$ and $j$, and the pair term ($\phi(r)$) and local electronic charge density function ($\psi(r)$) can be respectively expressed as:

$$\phi(r) = \begin{cases} A_1 \times exp\left[-p_1 \times \left(\frac{r}{r_0} - 1\right)\right], & r \leq r_{m1} \\ A_2 \times \left(\frac{r_{c1}}{r_0} - \frac{r}{r_0}\right)^{n_1} exp\left[-p_2 \times \left(\frac{r}{r_0} - 1\right)\right], & r_{m1} < r \leq r_{c1} \end{cases}, \tag{2}$$

$$\psi(r) = \begin{cases} B_1 \times exp\left[-q_1 \times \left(\frac{r}{r_0} - 1\right)\right], & r \leq r_{m2} \\ B_2 \times \left(\frac{r_{c2}}{r_0} - \frac{r}{r_0}\right)^{n_2} exp\left[-q_2 \times \left(\frac{r}{r_0} - 1\right)\right], & r_{m2} < r \leq r_{c2} \end{cases}, \tag{3}$$

where $r_{m1}$ and $r_{m2}$ are the knots; $r_{c1}$ and $r_{c2}$ are the cutoff radii for the pair and electronic charge density function terms, respectively; $r_0$ is a distance parameter; and $n_1 = 4$ and $n_2 = 5$ guarantee that $\phi$, $\psi$ and their derivatives can go to 0 at the cutoff radii smoothly. For the Al-Ni-Ti ternary system, there should be six sets of potential parameters, i.e., three sets of interactions for pure metals and three sets of cross interactions. In this work, the potential parameters were determined by fitting the calculated properties to their experimental or first-principle properties, such as lattice constants, cohesive energy, bulk modulus, elastic constants and vacancy formation energy. As a complement to experimental properties, first-principle calculations were carried out using the Vienna ab initio simulation Package (VASP) [27,28]. Verification and molecular dynamic simulations were carried out using the LAMMPS packages [29].

### 2.2. Atomic Simulation Models

In the present work, glass formation was depicted as a competition between the amorphous phase and its corresponding crystalline phase [7,13,30,31]. In addition, the energy difference between the amorphous and solid-solution phase served as the driving force of the crystalline-to-amorphous transition, and the driving force needed to be sufficiently large for amorphization to take place [12]. Therefore, the GFR was used to indicate the composition range within which the metallic glasses are energetically favored to form,

and the GFA was approximately related to the energy difference between the amorphous structure and its simple solid-solution structure. In order to investigate the GFR and GFA of the Al-Ni-Ti system, five sets of MD simulations were carried out at each composition point separated by 5 at% within the whole composition triangle. Interestingly, the exchange between atoms almost no longer affected the final energy for a certain structure when the atomic number was large enough. Therefore, there were $14 \times 14 \times 14 \times 4$ atoms and $19 \times 11 \times 12 \times 4$ atoms in fcc-based and hcp-based solid solutions, respectively. The solid solutions with different compositions were obtained by substituting heteroatoms in perfect crystalline lattices.

Then, MC simulations at 300 K and zero pressure were employed to structurally optimize the solid solutions to reach the state of minimum energy by optimizing their lattice constants and keeping the symmetry unchanged. After sufficient simulation time (approximately $1 \times 10^5$ steps when the energy fluctuation was less than 0.1 meV), the ideal solid solutions were obtained. When the energies of fcc- and hcp-based structures were different, the structure with lowest energy was chosen. Then, MD simulations were used to simulate the transitions from crystalline to amorphous. In the MD simulations, the ideal solid solutions were evolved at 300 K and 0 Pa for about $1 \times 10^6$ timesteps using the isothermal-isobaric (NPT) ensemble with a time step of $5 \times 10^{-15}$ s. A Nose–Hoover thermostat was applied, and hydrostatic pressure (iso) was used in the barostat. After sufficient evolution time, when the energy fluctuation was less than 1 meV/atom, the final structures generally evolved into the collapsed amorphous structures or remained as stable crystalline structures.

## 3. Results and Discussion

### 3.1. Construction of Al-Ni-Ti Interatomic Potential

For the Al-Ni-Ti ternary system, all the potential parameters were refitted with the TB-SMA formalism by Equations (1)−(3) in this work and are listed in Table 1. For pure Al, Ni and Ti metals, the fcc, bcc and hcp structures, respectively, and their corresponding physical properties were chosen as the fitting variables, as well as validation data. Similarly, the three most common experimental structures for Al-Ni, Al-Ti and Ni-Ti intermetallic compounds were chosen from the ICSD [32] database.

**Table 1.** Potential parameters of the TB-SMA model for the Al-Ni-Ti system.

|  | Al-Al | Ni-Ni | Ti-Ti | Al-Ni | Al-Ti | Ni-Ti |
|---|---|---|---|---|---|---|
| A1 (eV) | 0.4569 | 0.5586 | 0.3172 | 0.3564 | 1.6994 | 2.4855 |
| p1 | 7.5838 | 11.5942 | 9.9892 | 8.0628 | 7.4878 | 5.1036 |
| A2 (eV) | 0.4854 | 1.3281 | 2.0865 | 0.7006 | 0.0694 | 0.9432 |
| p2 | 2.7347 | 1.5300 | 0.1108 | 2.0128 | 2.2728 | 0.7224 |
| n1 | 4 | 4 | 4 | 4 | 4 | 4 |
| rm1 (Å) | 3.2824 | 2.9785 | 3.4976 | 3.3593 | 4.7839 | 3.0663 |
| rc1 (Å) | 5.6445 | 3.9029 | 4.7025 | 5.2709 | 6.8037 | 5.2128 |
| B1 (eV2) | 4.9693 | 6.7236 | 6.3238 | 5.6417 | 22.9707 | 35.0220 |
| q1 | 4.8330 | 3.2568 | 3.9940 | 7.3234 | 6.5876 | 4.9950 |
| B2 (eV2) | 0.1363 | 0.5683 | 0.1181 | 0.4487 | 1.6521 | 0.2281 |
| q2 | *0.6337* | 0.1000 | 0.1993 | 0.6539 | 2.9662 | 1.3153 |
| n2 | 5 | 5 | 5 | 5 | 5 | 5 |
| rm2 (Å) | 4.7207 | 2.4520 | 5.0156 | 4.6133 | 3.3745 | 4.5877 |
| rc2 (Å) | 8.1302 | 6.1358 | 8.9364 | 6.7808 | 7.0103 | 7.7823 |
| r0 (Å) | 2.8635 | 2.3258 | 2.9757 | 2.8913 | 2.6333 | 2.3510 |

The experimental data and physical properties reproduced by the newly constructed potential are shown in Tables 2–4. The tables show that the static physical properties for pure elements and their compounds are almost completely consistent with experimental data. To further compare the constructed potentials with the existing reported potentials, almost all the Al-Ni-Ti-related interatomic potentials since 2000 were selected to reproduce

the basic physical properties via LAMMPS and are listed in the Supplementary Material. Accurate predictions of the formation energies and energy sequences of pure Al, Ni, Ti elements and their various compounds, as well as reasonable ranges of binary metallic glass formation, are important criteria for an interatomic potential to predict ternary metallic glass formation. According to comparison, the constructed Al-Ni-Ti potentials appeared to be at a good level, so they were considered reliable to reproduce the physical properties of pure elements and their various compounds in this work.

**Table 2.** Properties of Al, Ni and Ti used for fitting and the reproduced results: lattice constants ($a_0$ and $c_0$), cohesive energy ($E_c$), bulk modulus ($B_0$) and elastic constants ($C_{ij}$). [ab] indicates that the data were calculated by first-principle calculations in this work.

| | Al | | | Ni | | | Ti | | |
|---|---|---|---|---|---|---|---|---|---|
| | **fcc** | **bcc** | **hcp** | **fcc** | **bcc** | **hcp** | **hcp** | **bcc** | **fcc** |
| $a_0$ (Å) | 4.05 | 3.22 | 2.85 | 3.52 | 2.81 | 2.48 | 2.95 | 3.28 | 4.13 |
| | 4.05 [33] | 3.23 [ab] | 2.85 [ab] | 3.52 [33] | 2.79 [ab] | 2.47 [ab] | 2.95 [33] | 3.25 [ab] | 4.10 [ab] |
| $c_0$ (Å) | - | - | 4.72 | - | - | 4.09 | 4.68 | - | - |
| | - | - | 4.73 [ab] | - | - | 4.09 [ab] | 4.68 [33] | - | - |
| $E_c$ (eV) | 3.39 | 3.37 | 3.39 | 4.44 | 4.39 | 4.44 | 4.85 | 4.82 | 4.85 |
| | 3.39 [34] | 3.29 [ab] | 3.36 [ab] | 4.44 [34] | 4.39 [ab] | 4.41 [ab] | 4.85 [34] | 4.74 [ab] | 4.79 [ab] |
| $B_0$ (GPa) | 72.2 | 70.7 | 72.0 | 189.6 | 177.4 | 189.6 | 106.4 | 102.1 | 106.2 |
| | 72.2 [34] | 68.4 [ab] | 74.4 [ab] | 186.0 [34] | 200.9 [ab] | 203.2 [ab] | 105.1 [34] | 109.6 [ab] | 111.5 [ab] |
| $C_{11}$ (GPa) | 83.1 | 70.6 | 97.6 | 242.5 | 162.7 | 303.1 | 131.2 | 81.8 | 169.8 |
| | 106.8 [33] | 41.3 [ab] | 106.1 [ab] | 248.1 [33] | 178.5 [ab] | 300.5 [ab] | 162.4 [33] | 98.6 [ab] | 140.0 [ab] |
| $C_{12}$ (GPa) | 66.7 | 70.8 | 66.2 | 163.1 | 184.8 | 158.4 | 94.0 | 112.2 | 74.5 |
| | 60.4 [33] | 81.9 [ab] | 67.8 [ab] | 154.9 [33] | 212.0 [ab] | 180.1 [ab] | 92.0 [33] | 115.0 [ab] | 97.2 [ab] |
| $C_{44}$ (GPa) | 29.1 | 33.2 | 15.0 | 123.4 | 121.0 | 67.6 | 61.9 | 97.8 | 33.8 |
| | 28.3 [33] | 41.3 [ab] | 6.5 [ab] | 124.2 [33] | 147.6 [ab] | 48.9 [ab] | 46.7 [33] | 45.4 [ab] | 59.2 [ab] |

**Table 3.** Properties of Al-Ni, Al-Ti and Ni-Ti compounds used for fitting and the reproduced results: lattice constants ($a_0$, $b_0$ and $c_0$), formation energy ($E_f$) and bulk modulus ($B_0$).

| | **Al$_3$Ni** | **AlNi** | **AlNi$_3$** | **Al$_3$Ti** | **AlTi** | **AlTi$_3$** | **Ni$_3$Ti** | **NiTi** | **NiTi$_2$** |
|---|---|---|---|---|---|---|---|---|---|
| | **D011** | **B2** | **L12** | **D022** | **L10** | **D019** | **hP16** | **B2** | **cF96** |
| $a_0$ (Å) | 6.56 | 2.84 | 3.51 | 3.86 | 2.82 | 5.77 | 5.10 | 2.96 | 11.31 |
| | 6.60 [35] | 2.88 [36] | 3.57 [37] | 3.85 [38] | 2.83 [39] | 5.78 [40] | 5.10 [41] | 3.01 [42] | 11.28 [43] |
| $b_0$ (Å) | 7.31 | - | - | - | - | - | - | - | - |
| | 7.35 [35] | - | - | - | - | - | - | - | - |
| $c_0$ (Å) | 4.77 | - | - | 8.59 | 4.05 | 4.64 | 8.32 | - | - |
| | 4.80 [35] | - | - | 8.58 [38] | 4.07 [39] | 4.65 [40] | 8.30 [41] | - | - |
| $E_f$ (eV) | −0.43 | −0.68 | −0.51 | −0.36 | −0.41 | −0.29 | −0.38 | −0.36 | −0.31 |
| | −0.42 [ab] | −0.69 [ab] | −0.47 [ab] | −0.38 [44] | −0.42 [44] | −0.26 [44] | −0.36 [45] | −0.35 [46] | −0.28 [45] |
| $B_0$ (GPa) | 92.6 | 152.9 | 156.4 | 71.5 | 136.8 | 112.1 | 130.8 | 123.4 | 83.4 |
| | 113.1 [ab] | 160.4 [ab] | 183.0 [ab] | 103.0 [47] | 112.1 [47] | 111.9 [47] | 163.4 [48] | 142.0 [45] | 119.8 [49] |

**Table 4.** Properties of Al-Ni-Ti compounds used for fitting and the reproduced results: lattice constants ($a_0$), cohesive energy ($E_c$), formation energy ($E_f$) and bulk modulus ($B_0$). lmp indicates that the data were calculated by LAMMPS, and MP indicates that the data were collected from the Materials Project.

| | **Pearson** | **Source** | **$a_0$ (Å)** | **$E_c$ (eV)** | **$E_f$ (eV)** | **$B_0$ (GPa)** |
|---|---|---|---|---|---|---|
| AlNi$_2$Ti | cF16 | lmp | 5.82 | 4.81 | −0.53 | 148.8 |
| | | MP [50] | 5.89 | 6.42 | −0.62 | 162.0 |
| Al$_{16}$Ni$_7$Ti$_6$ | cF116 | lmp | 12.09 | 4.44 | −0.50 | 78.7 |
| | | MP [51] | 11.80 | 5.66 | −0.56 | 130.0 |

In order to verify the reliability of the constructed Al-Ni-Ti potential, the equations of state (EOS) deduced from the TB-SMA potential are compared with the Rose equation. The simplified equation of Rose state is expressed as:

$$y = E_c \times \left(1 + \widetilde{a} + \delta \times \widetilde{a}^3\right) \times e^{-\widetilde{a}}, \tag{4}$$

$$\widetilde{a} = \sqrt{\frac{9 \times B_0 \times V_0}{E_c}} \times (x - 1), \tag{5}$$

where $x = a/a_0$, $a_0$ is the equilibrium lattice constant, $E_c$ is the cohesive energy, $B_0$ is the bulk modulus, $V_0$ is the equilibrium volume and $\delta$ is a fitted parameter.

Figure 1 shows the pair item (repulsive), local electronic charge density function term (attractive) and total energy as a function of lattice constants calculated from the TB-SMA potential and the corresponding Rose equations for fcc−Al, fcc−Ni, hcp−Ti, B2−AlNi, L10−AlTi and B2−NiTi. Figure 1 shows that the calculated total energy matches the Rose equation, and three parts of the TB-SMA potential are kept smooth in the whole calculation range. According to the above discussion, the constructed Al-Ni-Ti potential can reproduce the thermodynamic properties in agreement with the experimental observations, which confirms that the potential is reliable in describing the interactions of the Al-Ni-Ti system.

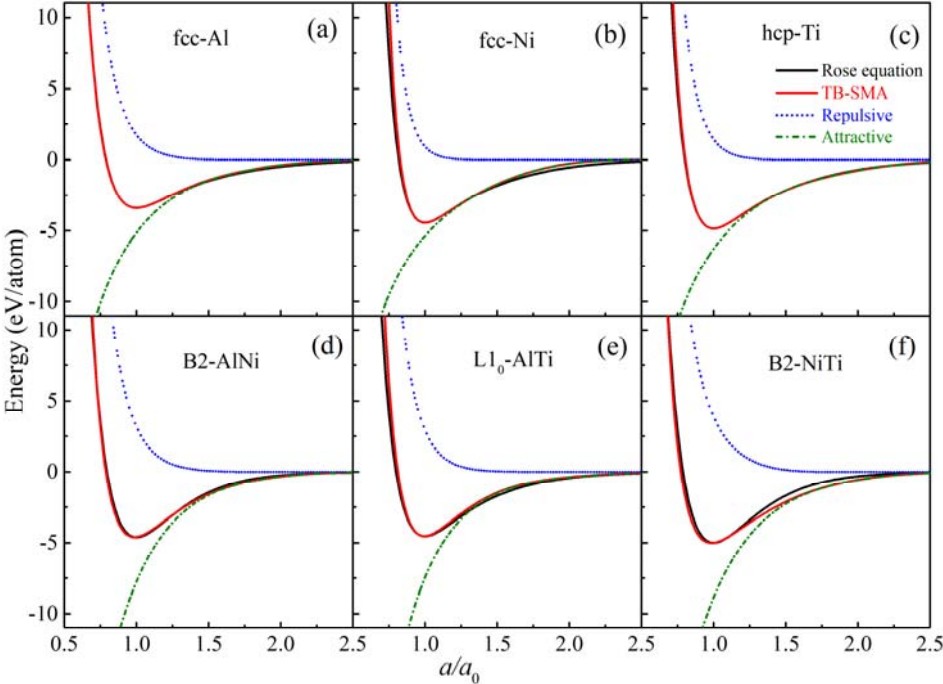

**Figure 1.** The equations of state (EOS) deduced from TB-SMA potential and the corresponding Rose equations for (**a**) fcc−Al, (**b**) fcc−Ni, (**c**) hcp−Ti, (**d**) B2−AlNi, (**e**) L10−AlTi and (**f**) B2−NiTi.

### 3.2. Glass-Formation Range and Ability of the Al-Ni-Ti System

Structural analysis was carried out based on the results of MD simulations. Figure 2 displays the pair correlation functions and projections of atoms in the $Al_{0.90}Ni_{0.05}Ti_{0.05}$ and $Al_{0.50}Ni_{0.25}Ti_{0.25}$ simulation models. Figure 2a shows that the $g(r)$ curve of $Al_{0.90}Ni_{0.05}Ti_{0.05}$ features discrete crystalline peaks, exhibiting a typical long-range ordered feature with the addition of small amounts off Ni and Ti in Al-based solid solution. Figure 2b shows that the $Al_{0.90}Ni_{0.05}Ti_{0.05}$ model preserves the original crystalline state. Figure 2c shows that beyond the second peak, all the peaks in $g(r)$ curves are smeared out and become indistinct. That is to say that the $Al_{0.50}Ni_{0.25}Ti_{0.25}$ model collapses and turns into a disordered state when the addition of Ni and Ti exceeds a critical value, as shown in Figure 2d. It is worth noting that a few simulation models showed a mixture of crystalline and amorphous phases

rather than a single phase, and the local structural environment of the models was further characterized.

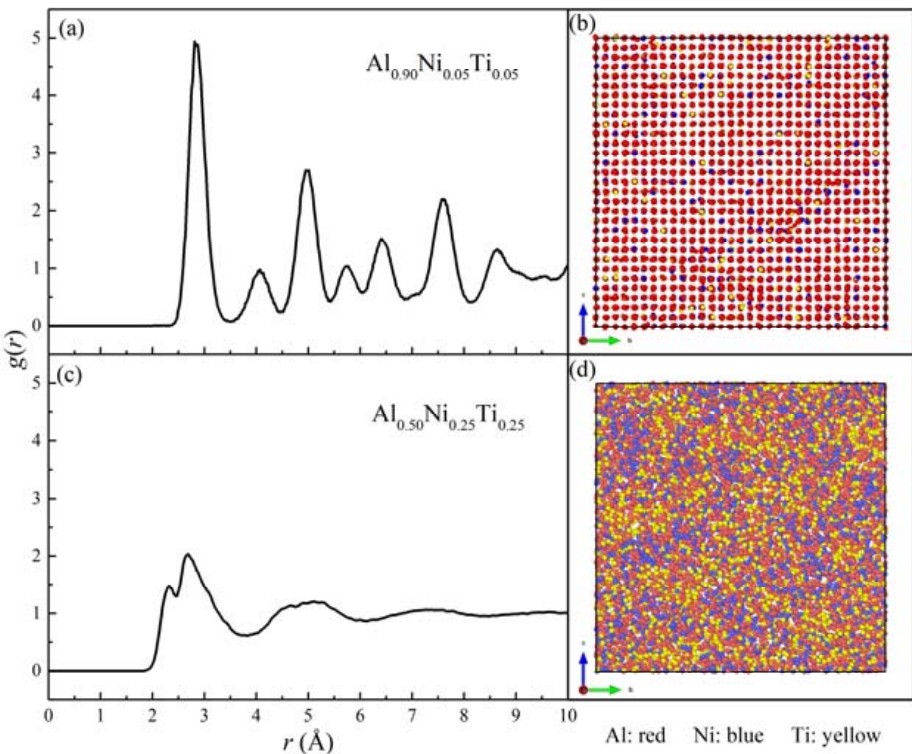

**Figure 2.** Total pair correlation functions ($g(r)$) and projections of atom positions for (**a,b**) the crystalline state ($Al_{0.90}Ni_{0.05}Ti_{0.05}$) and (**c,d**) the amorphous state ($Al_{0.50}Ni_{0.25}Ti_{0.25}$) obtained by MD simulations at 300 K. Red, blue and yellow filled circles stand indicated the positions of Al, Ni and Ti atoms, respectively.

A crystal-amorphous phase diagram was constructed according to the analysis over the entire Al-Ni-Ti composition triangle, as shown in Figure 3. Figure 3 shows that the Al-Ni-Ti system has a wide amorphous phase region, called the GFR, within which the alloys are energetically favored to form a disordered structure. The boundary regions between crystalline and amorphous phase regions are then roughly regarded as transitional regions. To validate the predicted GFR of the Al-Ni-Ti system, the identified experimental glass-formation compositions were extensively collected, as marked by colored dots in Figure 3. For instance, Kim [52], Liu [53], Lu [54] and Joress [15] have prepared metallic glasses by quenching onto a copper wheel, depositing on a Si wafer, ejecting onto a copper roller and cosputtering on wafers, respectively. Obviously, all the experimental compositions fall within the predicted GFR. Notably, the mixture structures of α-Al and amorphous phases prepared by Kim fall near the transition regions as well. Although Ni-based and Ti-based MGs are outstanding structural materials, few Ni-rich or Ti-rich ternary Al-Ni-Ti bulk MGs have been collected. In experiments, a variety of elements, such as Cu and Zr, are usually added to Al-Ni-Ti MGs to further improve the GFA. Therefore, there are a few experimental data available for pure ternary MGs. In general, external factors, such as impurities, chemical and structural inhomogeneities and producing process, may either stabilize or destabilize the solid solutions [12]. Therefore, the GFR predicted from the interatomic potential may vary from experimental observations.

Here, the larger the energy difference, the stronger the ADF for glass formation. After calculations of all target compositions, the amorphous driving forces for the Al-Ni-Ti system were calculated, as shown in Figure 4.

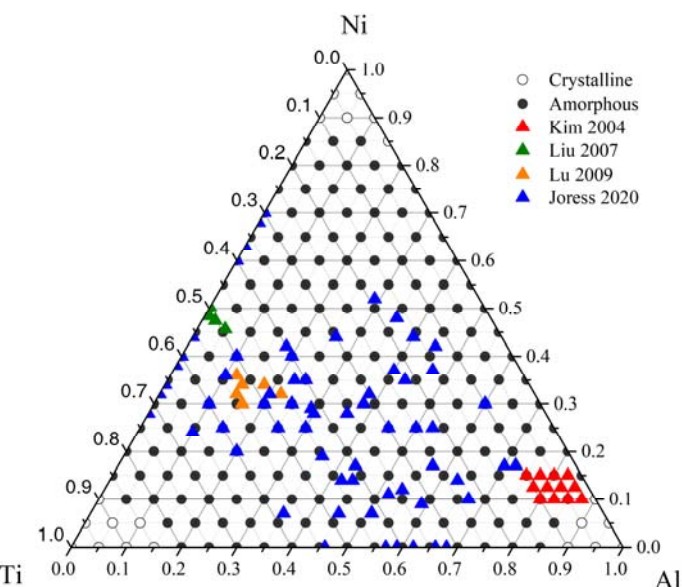

**Figure 3.** Crystal-amorphous phase diagram of the Al-Ni-Ti system derived from MD simulations at 300 K. Solid and hollow circles represent predicted states. Colored triangles represent experimental data from Kim [52], Liu [53], Lu [54] and Joress [15].

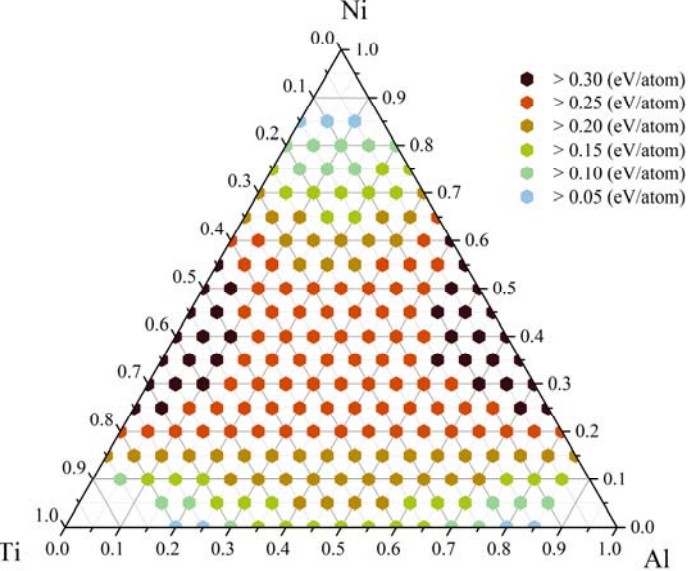

**Figure 4.** Amorphous driving forces for the transition of solid solutions to amorphous solutions in the Al-Ni-Ti system, with compositions located in the glass-formation region derived from MC and MD simulations.

To further evaluate the GFA of a specific Al-Ni-Ti alloy, the total energies of the $Al_xNi_yTi_{1-x-y}$ amorphous alloys and their corresponding solid solutions were calculated. The energy difference between the two states was defined as the amorphous driving force (ADF) for the crystal-amorphous transition, which could be a reliable indicator to reflect the ease or difficulty of glass formation for a specific alloy. The ADF for a specific composition can be expressed as:

$$\Delta E_{\text{driving force}} = E_{\text{solid solution}} - E_{\text{amorphous}}, \tag{6}$$

The energy difference is larger than 0.05 eV/atom over the whole GFR, which means that the formation of an amorphous phase is energetically favored. It is worth noting that the region with a large driving force (>0.15 eV/atom) covers most of the experimental

component points, and the regions with largest ADF (>0.30 eV/atom) appear near the $Ni_{0.40}Ti_{0.60}$, $Al_{0.65}Ni_{0.35}$ and $Al_{0.05}Ni_{0.35}Ti_{0.60}$ compositions. Interestingly, $Ni_{0.40}Ti_{0.60}$ falls in the medium between NiTi and $NiTi_2$ compounds in the Ni-Ti phase diagram, and $Al_{0.65}Ni_{0.35}$ falls in the medium between $Al_3Ni_2$ and $Al_3Ni$ compounds in the binary Al-Ni phase diagram. Just as Ni is a well-known alloying element with a significant effect on the enhancement of the GFA of metallic glasses [55–57], the addition of Ni into Al-Ti solid solutions results in a greater ADF for crystalline-to-amorphous transition until Ni becomes the main element of the solid solutions.

### 3.3. Local Atomic Arrangements of Al-Ni-Ti Metallic Glasses

In order to investigate the relationship between formation and structures of Al-Ni-Ti metallic glasses, the local atomic arrangements of $Al_{0.05}Ni_xTi_{0.95-x}$ alloys, abbreviated as $Ni_x$, were analyzed to explore the correlation between short- and medium-range order and glass-formation ability. As shown in Figure 4, $Ni_0$ and $Ni_{0.95}$ alloys remained as initial crystalline structures, whereas the structure of alloys collapsed when the concentration of Ni exceeded 10 at%. Among them, the $Ni_{0.35}$ alloy fell in the region with the largest amorphous driving force.

The pair-correlation function ($g(r)$) reflects the average distribution of near-neighbor distances. In the Figure 5, the $g(r)$ of $Ni_0$ and $Ni_{0.95}$ alloys show distinct crystalline peaks, which is consistent with the prediction of the GFR. When the content of Ni is higher than 10 at% and lower than 85 at%, the first two peaks become smooth, and the other peaks higher than 5 Å disappear, resulting in a short-range order and the long-range disorder in metallic glasses. In the amorphous structures, a new peak is formed, which is smaller than the distance of crystal $Ni_0$ and $Ni_{0.95}$ alloys, which may indicate that interstitial-like or low-coordination-number atomic configurations have been formed. Accordingly, the second peak shifts to a long distance relative to peaks in crystal structures, depending on the content of Ni and Ti in the alloy. Particularly in $Ni_{0.35}$ alloy, the intensities of the first and second peaks are similar. Figure 5b,c shows that the highest peak's intensity of $g(r)$ is related to the ADF of $Al_{0.05}Ni_xTi_{0.95-x}$ alloys, which means the larger the reciprocal of the highest peak's intensity of $g(r)$, the larger the ADF of the amorphous alloy.

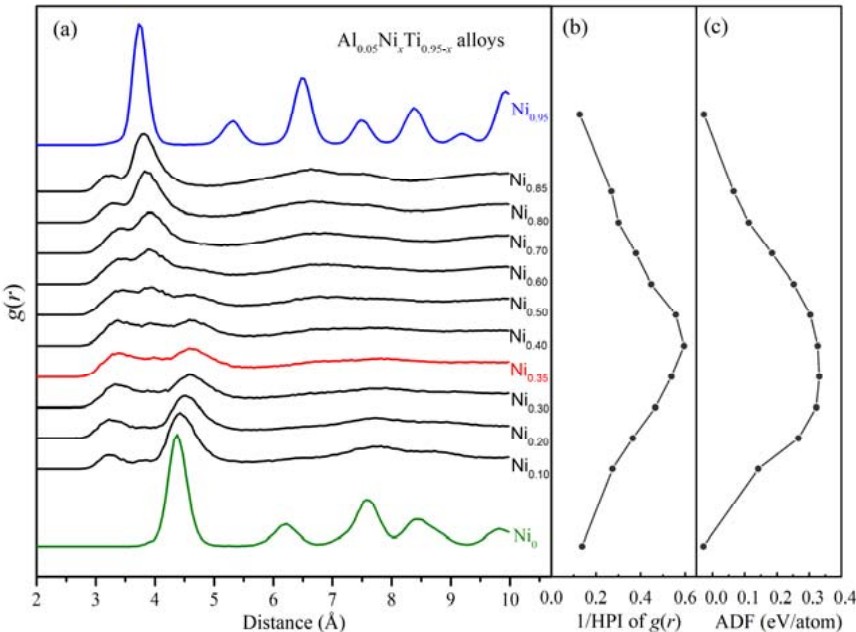

**Figure 5.** (**a**) Pair-correlation function ($g(r)$). (**b**) The reciprocal of the highest-peak's intensity (HPI) of $g(r)$. (**c**) The amorphous driving force (ADF) of the Al0.05NixTi0.95-x alloys obtained by MD simulations.

Apart from the g(r), the bond-angle distribution function (BADF) was further employed to reflect the orientation order of local atomic arrangements, as shown in Figure 6. In the crystal structures, the BADF both presents itself and a series of discrete peaks, such as 60°, 90°, 120° and 175. When the content of Ni exceeds 10 at% in hcp-Ti based alloy, the solid solution collapses, and new bond-angle peaks are formed, such as 86°, 130° and 167°. All the peaks become much wider and join together, indicating that new local atomic arrangements are formed in amorphous alloy, and the types of cluster configurations are more various. Especially in $Ni_{0.35}$ alloy, the whole curve of BADF becomes the flattest, indicating that the alloy has the most complex local atomic arrangements and the least crystalline features. Above all, the addition of Ni or Ti can increase both the disorder and complexity of the atomic arrangements of alloys, which is beneficial to the formation of metallic glasses.

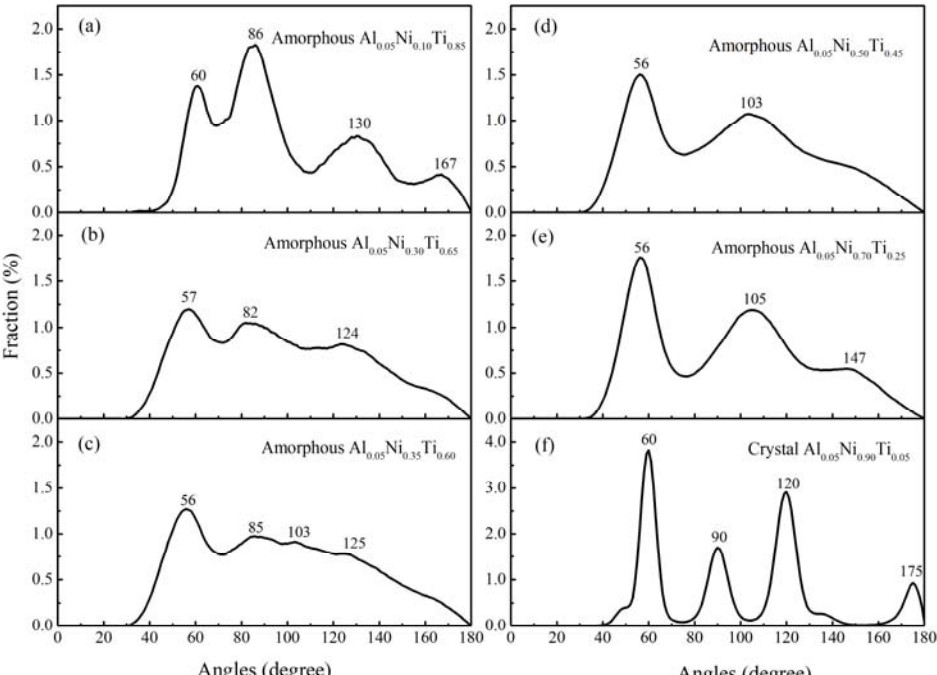

**Figure 6.** Total bond-angle distribution functions of alloys: (**a**) amorphous $Al_{0.05}Ni_{0.10}Ti_{0.85}$, (**b**) amorphous $Al_{0.05}Ni_{0.30}Ti_{0.65}$, (**c**) amorphous $Al_{0.05}Ni_{0.35}Ti_{0.60}$, (**d**) amorphous $Al_{0.05}Ni_{0.50}Ti_{0.45}$, (**e**) amorphous $Al_{0.05}Ni_{0.70}Ti_{0.25}$, (**f**) crystal $Al_{0.05}Ni_{0.90}Ti_{0.05}$.

The coordination number (CN) [58] is employed to characterize the average distributions of neighbor atoms. In the present work, the Voronoi analysis modifier was used in OVITO [59] software to calculate the Voronoi tessellation of the simulated alloys, which outputs the Voronoi indices and CNs. After tiny facets with an area less than 1% of the total area of the polyhedron surface are excluded [60], the CNs of crystalline alloys are computed as 12. With the addition of Ni, CNs of alloys become a distribution rather than a single number, as shown in Figure 7. The effect of minimal Ni addition to Ti-based alloys generates smaller and larger coordination numbers, such as 6 and 17, respectively. In $Ni_{0.35}$ alloy, the CN distribution is flattest, and the CNs of 13 and 14 are almost equal. Interestingly, all amorphous Nix alloys have almost the same average CN (12.7), which is different from that of crystalline alloys (12). When the content of Ni is increased as the main element in alloys, the median of CNs decreases from 14 to 13 for amorphous alloys. To some extent, the flatter the CNs, the more disordered the atomic arrangements of the alloys. Coincidentally, the $Ni_{0.35}$ alloy with the highest amorphous driving force has the flattest CN distribution.

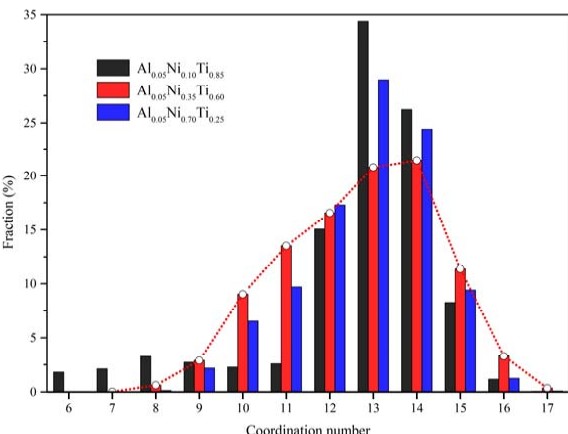

**Figure 7.** Distributions of the coordination numbers of the $Al_{0.05}Ni_xTi_{0.95-x}$ alloys. The dotted red curve represents the connection line of the distribution at $Al_{0.05}Ni_{0.35}Ti_{0.60}$ composition.

Moreover, Voronoi analysis and polyhedral template matching are employed to investigate the effect of element addition on the atomic arrangements [61–64].As shown in Figure 8, the dominant <0,2,8,5>, <0,2,8,4>, <0,2,8,2>, <0,1,10,2> and others in amorphous alloys are all the distorted icosahedral clusters with fivefold symmetry and abundant pentagonal faces, different from <0,2,4,6> in crystalline alloys. In Ti-rich alloys, the addition of Ni leads to the appearance of some new Ni-centered Voronoi polyhedrons with low CNs, such as <0,2,2,2>, <0,2,5,2>, <0,3,4,3>, which are much different from Ti- or Al-centered Voronoi polyhedrons. In contrast, adding Ti to Ni-rich alloys does not lead to the emergence of new Voronoi polyhedrons. In other words, Ti-centered Voronoi polyhedrons are prone to maintain the existing Ni-centered polyhedron types. In the $Al_{0.05}Ni_{0.10}Ti_{0.85}$ and $Al_{0.05}Ni_{0.70}Ti_{0.25}$ alloys, approximately regarded as adding Al in Ti-rich and Ni-rich amorphous alloys, respectively, the Al-centered Voronoi polyhedrons are also prone to maintaining the existing polyhedron types rather than generating a new one, which is similar to the effect of Ti addition. In light of this, the addition of Ni plays a more effective role in increasing structural disorder than Al and Ti, which is manifested as not only increased diversity of local atomic arrangements but also a reduction in the population of dominant Voronoi polyhedrons. In other words, the addition of Ni is more beneficial to the formation of metallic glasses, which is consistent with experimental observations [19,25].

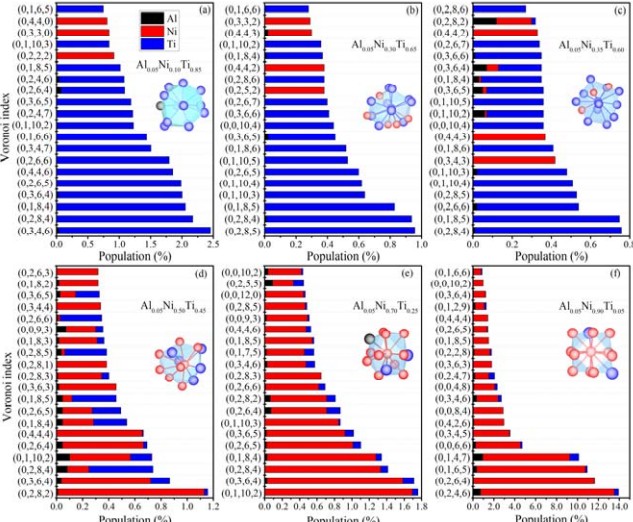

**Figure 8.** Populations of the most dominant Voronoi polyhedrons obtained from MD simulations and the coordination cluster with the highest population for alloys:(**a**) $Al_{0.05}Ni_{0.10}Ti_{0.85}$, (**b**) $Al_{0.05}Ni_{0.30}Ti_{0.65}$, (**c**) $Al_{0.05}Ni_{0.35}Ti_{0.60}$, (**d**) $Al_{0.05}Ni_{0.50}Ti_{0.45}$, (**e**) $Al_{0.05}Ni_{0.70}Ti_{0.25}$, (**f**) $Al_{0.05}Ni_{0.90}Ti_{0.05}$.

## 4. Conclusions

A long-range empirical potential was constructed for an Al-Ni-Ti system under the formalism of second-moment approximation of tight-binding potential. Accordingly, the glass-formation range (GFR) of the Al-Ni-Ti ternary system was determined by comparing the relative stability of the two competition phases. Then, the amorphous driving force (ADF) of alloys was obtained by calculating the energy difference between the two competition phases, which was regarded as an indicator of glass-formation ability (GFA) of alloys. The alloy around the $Al_{0.05}Ni_{0.35}Ti_{0.60}$ composition was most energetically favored to prepare metallic glasses. Furthermore, the local atomic arrangements of $Al_{0.05}Ni_xTi_{0.95-x}$ alloys were analyzed to elucidate the correlation between the atomic structure and ADF, revealing that the larger the ADF of an alloy, the falter its pair correlation function, bond-angle distribution function, coordination number distribution and Voronoi index distribution. Combining the ADF and structure analysis, the addition of Ni, less than 35 at% in $Al_{0.05}Ni_xTi_{0.95-x}$ alloys, could significantly increase the ADF, configurations of five-fold symmetry and structural disorder of alloys, which could improve the GFA of Al-Ni-Ti metallic glasses.

**Supplementary Materials:** The following supporting information can be downloaded at: https://www.mdpi.com/article/10.3390/cryst12081065/s1, Figure S1: Properties of Al reproduced by different interatomic potentials: lattice constant ($a0$ and $c0$), cohesive energy ($Ec$), bulk modulus ($B0$), elastic constants ($Cij$) and melting point ($Tm$). ab indicates that the data were calculated by first-principle calculations in this work; Table S2: Properties of Ni reproduced by different interatomic potentials; Table S3: Properties of Ti reproduced by different interatomic potentials; Table S4: Properties of Al-Ni reproduced by different interatomic potentials; Table S5: Properties of Al-Ti reproduced by different interatomic potentials; Table S6: Properties of Ni-Ti reproduced by different interatomic potentials; Table S7: Properties of Al-Ni-Ti reproduced by different interatomic potentials. References [65–90] are citied in the Supplementat Materials.

**Author Contributions:** Conceptualization, Q.Y., J.L. (Jiahao Li), W.L., J.L. (Jianbo Liu) and B.L.; data curation, Q.Y.; formal analysis, Q.Y.; funding acquisition, J.L. (Jiahao Li), W.L., J.L. (Jianbo Liu) and B.L.; investigation, Q.Y.; methodology, Q.Y., J.L. (Jiahao Li) and W.L.; project administration, J.L. (Jianbo Liu) and B.L.; resources, J.L. (Jianbo Liu) and B.L.; software, Q.Y. and J.L. (Jiahao Li); supervision, B.L.; validation, J.L. (Jiahao Li), W.L. and J.L. (Jianbo Liu); visualization, Q.Y.; writing—original draft, Q.Y.; writing—review and editing, Q.Y. and J.L. (Jianbo Liu). All authors have read and agreed to the published version of the manuscript.

**Funding:** This research was funded by the National Natural Science Foundation of China (grant number 51631005) and the Ministry of Science and Technology of the People's Republic of China (grant number 2017YFB0702201).

**Institutional Review Board Statement:** Not applicable.

**Informed Consent Statement:** Not applicable.

**Acknowledgments:** The authors are grateful for the financial support from the Key Laboratory of Advanced Materials and the Administration of Tsinghua University.

**Conflicts of Interest:** The authors declare no conflict of interest. The funders had no role in the design of the study; in the collection, analyses, or interpretation of data; in the writing of the manuscript; or in the decision to publish the results.

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
