# Peer review of "Interatomic Potential to Predict the Favored Glass-Formation Compositions and Local Atomic Arrangements of Ternary Al-Ni-Ti Metallic Glasses"

_crystals, doi:10.3390/cryst12081065_

Round 1

Reviewer 1 Report

Authors present results from extensive simulations using an appropriate potential for the interatomic interaction on the phase behavior of ternary Al-Ni-Ti systems.

The work addresses an important topic from a modelling/simulation perspective, it is well explained (in general, see comments below) and well documented. One could claim that glass formation is the antithesis of the physicochemical concepts covered in “Crystals” but the present work can be of interest to a general readership.

There are some problems with the syntax and grammar and there are some parts of the methodology which are quite unclear.

Subject to addressing the comments and questions below the work will be suitable for publication in Crystals.

) Parts of the methodology are unclear. Information should be added as the simulations should be reproducible by independent researchers. For example, how the MC simulations are performed? What kind of displacement is allowed? How the different species are treated? Do the authors use their own code of this is done through the software programs mentioned earlier?

“after sufficient simulation time about 1x10^5 steps”. How authors decide on the sufficiency of the simulation time? How they track the evolution of the relaxation of the system? Is this related to the energy fluctuations mentioned later?

Additional details should be used on the thermostat and barostat used in MD. Also, in which ensemble the MC simulations are conducted?

) Figure 3. Authors should remove the names from the figure legend and replace them with the references or modify them so that they further include the corresponding references. As they are now they seems very awkward. Also, experimental data should be colored, as done by the authors, but they should be also circles or of a specific shape for all experimental data. As it is now the coloring and the use of different symbols is confusing especially since in the legend it is reported “colored dots” for the experimental data.   

) Authors need to check carefully and edit accordingly their manuscript for syntax and grammar errors. In numerous instances use of plural / singular is not correct.

) What is meant by “and their species are various” ? (line 242)

) Dotted red curve should be properly explained in Fig. 7.

) Line 102-104 should be removed.

) Figure 4, legend. “Amorphous .. transition” -> “Driving forces … transition of solid solutions to amorphous”.

) Authors should include in their reference list recent relevant tools for the structural analysis of similar computer-generated configurations. See for example Modelling Simul. Mater. Sci. Eng. 24, 055007 (2016); Commun. Matter 1, 1 (2020); Crystals 10, 1008 (2020)]

Reviewer 2 Report

1.      The authors have not discussed the existence of <0, 0, 12, 0> clusters which are the structural motif in metallic glasses.

2.      The method of obtaining glassy structure is unclear, and there is no suitable reference to the method employed (section 2.2)

3.      The number of atoms mentioned in section 2.2 is unclear. Is it the simulation box size? What are the dimensions along x-, y-, and z-axes?

4.      The authors could have done mechanical testing (tensile/compression) to evaluate the mechanical properties and compare the results with the studies reported using a different Al-Ni-Ti potential.

1.      The authors can also comment/compare the computational time for predicting the properties with reference to the other available interatomic potentials. 

Reviewer 3 Report

Dear editor,

In the manuscript, the authors construct an empirical potential by modifying the Second Moment Approximation (SMA) formalism of the original Tight Binding (TB) potential for the Al-Ni-Ti ternary system. To test the accuracy of the new potential, they tested the physical properties of pure Al, N, and Ti elements and their various compounds with the LAMMPs package program. It is clear that the authors put a lot of time and effort into the work, and I appreciate that. However, I think the manuscript needs major revision before it can be evaluated. Below I list my observations, missing points, concerns, and recommendations about the manuscript.

1.       First, I wait for the authors to explain why they need such parameterization for the ternary Al-Ni-Ti alloy, and I think this should be added to the manuscript as well. Because there are many potentials that can be used for these materials in the literature. Even, the authors have included these potentials in detail in the supplementary materials section of the manuscript.

2.       Another point is why did the authors go to such trouble and not use the TB-SMA potential parameters reported by Cleri and Rosato [1] for fcc and hcp metals? I think they should definitely clarify this issue. Then readers will better understand why this new regulation is necessary. For example, in some studies previously reported using TB-SMA potentials, it was reported that the results were not very good for Al [2]. By reviewing this and similar works, authors can discuss their justification for reparameterization in their manuscripts in more detail.

3.       The authors presented the test results of the new parameters for low temperatures or solid properties of materials. However, for the reliability of the results, it is very important that the potentials are transferable. For this, energy-temperature curves and naturally the determination of melting points of pure Al, Ni and Ti elements during the heating process are needed. Melting temperatures are given for other potentials in additional materials, but I could not find data for this new potential. In addition, it is vital to compare the pair distribution functions and structure factors with the experimental results for liquid Al, Ti, and Ni.

4.       Another thing I'm curious about is that as far as I know, the TB-SMA potential is not in the LAMMPS package program (at least for metallic systems). I think it would be very helpful to inform the readers about this subject.

If the authors give reasonable explanations about the issues I mentioned above, then I think the manuscript can be reconsidered.

 [1]         Phys. Rev. B. 48 (1993) 22–33. https://doi.org/10.1103/PhysRevB.48.22.

[2]         Intermetallics. 84 (2017) 62–73. https://doi.org/10.1016/j.intermet.2017.01.001.

Reviewer 4 Report

The paper is devoted to the computational study of crystal/glass transition in an AlNiTi alloy. The novel interatomic potential was developed. The glass-formation ability was studied for a wide range of AlNiTi compositions. A thoughtful structural analysis was performed. Overall, in my opinion, this manuscript can be published in Crystals after minor revision.

Questions and comments.

1. Page 1. "1959 [4]". Ref 4 refers to 1960.
2. Page 1. "In experiments, the GFA of MGs is evaluated to be proportional to either its critical cooling rate or casting size".
The references supporting this statement should be given.
3. Page 1. "...demonstrate the correlation of the full width at half maximum in X-ray diffraction...". This correlation is unclear and should be explained.
4. Page 2. "...Al-rich MGs exhibit obvious plasticity...". It does not seem obvious for an external reader and should be clarified.
5. Page 1. "Liu et al ... [11-13]". Liu was not the author of Refs 11-13.
6. Page 2. "In this work, the potential parameters were determined by fitting the calculated properties to their experimental or first-principle properties." These properties should be explicitly listed.
7. Page 2. "a large number of MD simulations were carried". The number of performed simulations should be specified.
8. The interatomic potential was developed for a crystalline state. How good can it describe the liquid properties?
9. Table 3. Why "B2, L1_2, ..." and other structures were chosen? They are most stable in experiments? Is it proven they are stable with this new potential?
10. Table 4. The abbreviations "lmp" and "MP" are not clear.
11. Page 8. " Just as observed in experiments". The references should be given.
12. Page 11. "4. Discussion" should be changed by "4. Conclusion".
13. The following typos were found. Page 1, "In Recent years" -> "In recent years"; Page 3, "atom number" -> "atomic number"; Page 3, "finial" -> "final"; Page 10, "Nix" should be in subscript

Round 2

Reviewer 3 Report

 Dear editor,

The authors gave mostly convincing answers to my questions. In general, I can say that they have resolved some of my concerns. However, for the necessity of the new parameterization, I strongly suggest that they take support from the sources in the literature and cite them as a source. This will also make the work of researchers easier and I believe that it will attract the attention of a larger audience. Especially for pure Al element, proper parameterization is really important in MD simulations. Therefore, as I said before, I strongly recommend scanning the relevant articles in the literature and adding an explanation sentence to the introduction [1]. On the other hand, TB-SMA potentials are among the most preferred potentials for MD simulation calculations. It is also commendable that the authors have adapted this potential form to the LAMMPS package program. I would like to let you know that I would be glad if they share their implementation in a forum or through you to embed the TB-SMA potential form in LAMMPS.

In conclusion, I recommend that the manuscript be published in your journal.

[1] Intermetallics. 84 (2017) 62–73. https://doi.org/10.1016/j.intermet.2017.01.001.

This manuscript is a resubmission of an earlier submission. The following is a list of the peer review reports and author responses from that submission.

Round 1

Reviewer 1 Report

Recommendation: Reject (see attached comments). I am not willing to spend time on any further review.

Reviewer 2 Report

The paper is an  interesting theoretical work on the prediction of glass forming ability of Al-Ti-Ni metallic glasses. The authors present a tedious MD simulation by setting the parameters  of a selected atomic potentials and predicting the stability and some physical parameters of known binary and some ternary compounds. They compare the total energy of the solid solution and the formed nonperiodic structure and describe the variation of short range order and local atomic characteristics for the whole ternary phase diagram. Altogether it is a valuable piece of work. I suggest to accept it after the revision of the following minor points. 

The long sentence in the middle of page 3 starting with "To ensure ..." is and the subsequent sentence are confusing. Please, rewrite this part to make clear for the reader.

Please, indicate the unit (eV/atom) at the color scale of Fig 4.

Reviewer 3 Report

The paper represents computer simulation of metallic glass formation in Al-Ni-Ti alloys. Although, the results may be of interest for computational community they are hardly acceptable as physically possible. Solid state amorphization at 300 K is completely unusual. It is not expected at all, because real atomic diffusion at such temperature is too slow! Please, estimate the diffusion coefficients (and the diffusion length on laboratory timescale) from the higher temperature data published in Smithells Metals Reference Book, for example, or any other literature source. If the authors really need to determine the glass-formation ability, then a series of computations on cooling from the liquid state at different cooling rates shall be made.

Moreover, a clearly inhomogeneous structure is seen in Fig. 2(d). Is it phase separation?

In view of the above-mentioned the paper shall be rejected.

Round 2

Reviewer 3 Report

I am still not convinced with the approach used by the authors even they are likely experts in the field of computational research.

Room-temperature solid state amorphization is declared to take place at room temperature which is rather not possible: “In the MD simulations, the ideal solid solutions were evolved at 300 K and 0 Pa for about 1×10^6 timesteps, using the isothermal-isobaric (NPT) ensemble with a time step 5×10-15 s. After sufficient evolution time when the energy fluctuation was less than 1 meV/atom, the finial structures would generally evolve into the collapsed amorphous structures or remain the stable crystalline structures”.

Moreover, most likely so heavily alloyed solid solutions (which disintegrate at room temperature) could hardly be formed in principle! Then, there is no physical meaning in studying them. Computer modeling can do many things, some of which are just unrealistic, however.

The authors reply: “As the process of making metallic glasses is always far from equilibrium process, the complicated phase cannot nuclear (nucleate?) and grow due to the kinetic condition. Therefore, the competing phase of metallic glasses is the terminal solid solution, which structure is relatively simple (Liu, et al. Applied Physics Letters 42.1 (1983))”.

- When ternary metallic glasses are made by cooling liquids intermetallic compounds (not solid solutions) are usually formed because the compositions are close to the center of compositional triangles. Terminal solid solutions in Al-Ni-Ti system can be formed only in narrow areas close to the corners of the phase diagram. In order to study real glass-forming ability liquid cooling processes must be modeled.

However, if the authors model sold-state amorphization at room temperature by severe plastic deformation such as ball milling, then it is well known that the compositional regions for amorphous phase in such case are significantly different from those obtained by rapid cooling of a liquid. Here amorphization takes place close to the intermetallic compound compositions rather then at eutectics.

Clear chemical inhomogeneity in Fig. 2 is still an issue.